# Genetic Variants and Lifestyle Factors in Androgenetic Alopecia Patients: A Case–Control Study of Single Nucleotide Polymorphisms and Their Contribution to Baldness Risk

**DOI:** 10.3390/nu17020299

**Published:** 2025-01-15

**Authors:** Roberto Ambra, Simona Mastroeni, Sonia Manca, Thomas J. Mannooranparampil, Fabio Virgili, Barbara Marzani, Daniela Pinto, Cristina Fortes

**Affiliations:** 1Research Centre for Food and Nutrition, Council for Agricultural Research and Economics (CREA-AN), 00178 Rome, Italy; sonia.manca84@gmail.com; 2National Centre for Disease Prevention and Health Promotion, Italian National Health Institute, 00161 Rome, Italy; simona.mastroeni@iss.it; 3Dermatology Unit, Salus Infirmorum Clinic, 00135 Rome, Italy; tjmannooran@gmail.com; 4Interuniversitary Consortium “National Institute for Bio-Structures and Bio-Systems” (INBB), 00165 Rome, Italy; fvirgili@outlook.it; 5R&D Giuliani S.p.A., 20129 Milan, Italy; bmarzani@giulianipharma.com (B.M.); dpinto@giulianipharma.com (D.P.); 6Epidemiology Unit, Istituto Dermopatico dell’Immacolata (IDI-IRCCS-FLMM), 00167 Rome, Italy; c.deppermannfortes@idi.it

**Keywords:** androgenetic alopecia, single nucleotide polymorphisms, diet

## Abstract

Single nucleotide polymorphisms (SNPs) found to be associated with Androgenetic Alopecia (AGA) to date, are characterized by an apparent reduced penetrance into the phenotype suggesting a role of other factors in the etiology of AGA. Objective: We conducted a study to investigate the role of specific allelic variants in AGA controlling for nutritional and lifestyle factors. Methods: Individual patterns of SNPs present in the baldness susceptibility locus at 20p11 (rs1160312 and rs6113491) or close to the androgen receptor (AR) gene in chromosome X (rs1041668) were investigated in 212 male subjects. Information on socio-demographic characteristics, medical history, smoking, and diet was also collected. Logistic regression was used to estimate odds ratios (ORs) and 95% confidence intervals (CIs). Results: After controlling for age, diet, BMI, family history of AGA, and smoking, an increased risk of AGA was found for subjects with [A] alleles for both rs1160312 (OR: 2.97; 95% CI: 1.34–6.62) and rs6113491 (OR: 2.99; 95% CI: 1.37–6.52), and for subjects with the TT genotype for rs1041668 (OR: 4.47; 95% CI: 1.60–12.5). Multivariate logistic regression indicates that diet, familiarity, and BMI, but not smoking, remain statistically significant despite the different SNP genotypes. Conclusions: To our knowledge, this is the first indication that the rs1160312, rs6113491, and rs1041668 polymorphisms are independent risk factors for AGA that can be modulated by diet.

## 1. Introduction

Androgenetic Alopecia (AGA) is a sex-limited and age-related dermatological condition that affects around 80% of European men and is characterized by the severe shrinking of hair follicles and eventually by evident hair loss [1]. The proportion of affected males is strongly influenced by race/ethnicity, but shows a worldwide increasing trend [2]. Epidemiological studies have shown that AGA is associated with an increased risk of many chronic diseases, including cardiovascular diseases, diabetes, and prostate cancer [3], suggesting common pathophysiological mechanisms and/or nutritional and environmental factors.

Early studies on twins attributed ≈80% of the phenotypical variance of AGA to additive genetic factors [4]. Accordingly, genome-wide association studies (GWASs) conducted over the subsequent two decades identified more than 350 risk genomic regions and more than 600 independent, common, single nucleotide polymorphisms (SNPs) [5]. However, a recent, large, pedigree-based heritability analysis reduced the contribution of genetics and SNPs to AGA phenotypic variance at 60% and 39%, respectively [6], supporting the role of other factors in the etiology of AGA, including lifestyle and nutritional factors. In fact, hair function and appearance have been reported to rely largely on an adequate and balanced nutritional intake [7], with recent evidence supporting the protective role of the phytochemicals of nutritional plants typical of the Mediterranean diet, naturally rich in polyphenols and antioxidants [8,9,10].

From this perspective, we conducted a study aimed at investigating the relationships between previously associated SNPs and environmental factors in determining the risk of AGA. For such a purpose, individual patterns of genetic variants strongly associated with AGA located in the baldness susceptibility loci at 20p11 (rs1160312 and rs6113491) [11,12] or in chromosome X between the AR and the ecto-dysplasin A2 receptor (EDA2R) genes (rs1041668) [13] were compared to the individual dietary intake and smoking status of 104 males affected by AGA and 108 controls.

## 2. Materials and Methods

### 2.1. Subjects and Food Items

This case–control study was conducted as part of a study that investigated the possible risk factors for the development and the severity of AGA [9,14]. Briefly, males affected by AGA (N = 104) of European ethnicity aged 18 years or more and resident in the Lazio region were drawn from out-patient clinics of the hospital “Istituto Dermopatico dell’Immacolata” (IDI), the reference hospital for dermatological diseases in the Lazio region, between 2011 and 2012. Males not affected by AGA (N = 108) of European ethnicity aged 18 years or older, resident in the Lazio region, were chosen as controls. To avoid bias, both cases and controls were drawn from the same geographic region. A trained dermatologist collected information on socio-demographic and clinical characteristics, including weight and height, the presence of chronic diseases (e.g., diabetes, dyslipidemia, cancer, cerebrovascular, and cardiovascular diseases), family history of AGA, smoking, and dietary habits. The trained dermatologist, with the help of 12 series of figures, observed the subject’s head from two angles (side and top) and classified the hair pattern according to the Hamilton baldness scale, as modified by Norwood [15].

Dietary intake was assessed by a validated food-frequency questionnaire [16,17]. The consumption of food groups was defined on a seven-point scale and then combined in two categories according to the frequency distribution among the controls. Food items were grouped on the basis of rough phytochemicals’ content. For example, spinach, chicory, and beet leaves formed the “dark leafy green vegetables” (a good source of phenols, lutein, and zeaxanthin) and broccoli, cauliflower, and cabbage formed the “cruciferous vegetables” (good sources of isothiocyanates and indoles). The use of fresh herbs, such as parsley, sage, basil, and rosemary, were categorized as the “number of fresh herbs regularly used”. In the latter case, low intake was defined as the regular consumption of two or less fresh herbs, while the regular consumption of three or more herbs was considered a high intake. Smoking status was categorized into two groups (ever-smoker/non-smoker).

Body mass index was calculated as weight in kilograms divided by height in meters squared. Family history was defined as having a first-degree relative affected by AGA.

All participants in the study who filled in the questionnaire were asked to provide a saliva sample. DNA was obtained from 212 subjects. Individual patterns of SNPs located in chromosome 20 (rs1160312 and rs6113491) or in chromosome X between the AR and the ectodysplasin A2 receptor (EDA2R) genes (rs1041668) were investigated in all 212 subjects. The study was approved by the IDI ethical committee and written consent was obtained from all participants.

### 2.2. SNP Genotyping

DNA was extracted from saliva samples using the Puregene DNA purification kit (DNA Genotek, Kanata, Ontario, Canada), slightly modifying the manufacturer’s instructions. Briefly, 500 µL of the saliva sample mixed with the Oragene DNA Stabilizing solution was transferred in a 2 mL reaction tube and incubated for 1 h at 50 °C, with shaking the tube every 5 min. A total of 125 µL of a cell lysis solution and 3 µL of RNase A were added. After incubation for 10 min at RT, 200 µL of a protein precipitation solution was added, and the tubes were centrifuged at 10,000 rpm for 10 min. One volume of isopropanol and 5 µL of glycogen were added, and the tubes were centrifuged at 13,000 rpm for 5 min. The supernatant was discarded, and the pellets washed with 70% ethanol and then centrifuged again at 13,000 rpm for 5 min. Finally, the pellets were resuspended in 50 µL of a DNA hydration solution. Genomic DNA was quantified using a Nanodrop ND-1000 instrument (Agilent technologies, Santa Clara, CA, USA) and diluted to a final concentration of 5 µg/µL.

A Tecan Freedom EVO75 robot was utilized to handle and load genomic DNA (10 ng) and Master mixes in 384-well PCR plates. Genotyping was performed using TaqMan assays specific for every SNP, run on an ABI Prism 7900 HT Sequence Detection system using the TaqMan^®^ Universal PCR Master mix (Applied Biosystems, Foster City, CA, USA). The thermal cycling conditions were: pre-PCR read for the fluorescence background; PCR cycling: 95 °C for 10 min for activating AmpliTaq Gold; 40 cycles at 92 °C for 15 s for denaturing genomic DNA and 1 min at 60 °C for annealing/extension; and finally, a post-PCR read for fluorescence quantification at the end of the run. Standard curves using serial dilutions of known concentrations of genomic DNA were performed to verify the assays’ sensitivity, and technical replicates were performed for each sample to assess the consistency of the results. Both the positive controls (known genotypes) and negative controls (no template controls) were included in each run to verify the assays’ specificity and detect possible contamination. For each SNP, a raw data table was created, and the major and minor allele frequencies for each sample were subjected to statistical analysis.

### 2.3. Statistical Analysis

Unconditional logistic regression was used to investigate the association between SNPs and AGA in the baldness susceptibility locus at 20p11 (rs1160312 and rs6113491) or near the AR gene (rs1041668) located in chromosome X. Odds ratios (ORs) and 95% confidence intervals (CIs) for the intermediate- and high-exposure categories were calculated using as reference categories the TC genotype for rs1041668 and the GG and CC homozygotes for rs1160312 and rs6113491, respectively, and pooling the AA and AG genotypes for rs1160312 and the AA and AC ones for rs6113491. Different multivariate logistic regression models controlling for possible confounders were run. We included, in the multivariable models, only variables that were statistically significant in the univariate analysis (*p*-value < 0.05). The following variables were considered in the same multivariate regression model, to adjust for each other as potential confounders: age, BMI, family history of AGA, smoking, and food items previously shown to be associated with AGA. All analyses were performed using the statistical software package PC-STATA (Stata Statistical Software: Release 15. College Station, TX, USA: StataCorp LLC).

## 3. Results

Table 1 shows the characteristics of the subjects participating in the study. The mean age of the participants was 28.5 years old for the cases and 38.9 years old for the controls. Out of 104 AGA subjects, 35 (37.5%) had a moderate/severe AGA. Age, BMI, family history of AGA (*p*-value < 0.001), and smoking (*p* = 0.05) were all associated with AGA. No difference was found for education and the presence of chronic diseases. AGA subjects had a higher frequency of allele A in rs1160312 (76.0% vs. 57.4%) and rs6113491 (75.0% vs. 58.3%) SNPs in comparison to the controls. Also, the TT genotype of rs1041668 was more frequent in AGA subjects than in the controls (91.3% vs. 74.1%).

Table 2 shows the crude and adjusted risk estimates for rs1160312, rs6113491, and rs1041668. After the adjustments for age, BMI, family history of AGA, ever-smoking, and consumption of salad and fresh herbs, an increased risk was found for the TT genotype of rs1041668 near the AR gene locus (OR: 4.47; 95% CI: 1.60–12.5) and for subjects carrying allele A of rs1160312 (OR: 2.97; 95% CI: 1.34–6.62) and rs6113491 (OR: 2.99; 95% CI: 1.37–6.52). In the multivariate models, the consumption of salad, the use of fresh herbs, familiarity, and BMI remained statistically significant, while smoking was no longer statistically significant (in model 2, for those who have quit smoking and current smokers vs. never having smoked, OR: 0.61; 95% CI: 0.30–1.25, OR: 0.58; 95% CI: 0.29–1.17, and OR: 0.60; 95% CI: 0.30–1.20 for rs1160312, rs1041668, and rs6113491, respectively).

## 4. Discussion

Our study shows that genetic variants located in the baldness susceptibility loci at 20p11 or in chromosome X between the AR and the ectodysplasin A2 receptor (EDA2R) genes are associated with an increased risk of AGA after controlling for all possible confounders. Our observations confirm the early associations of rs1041668 [13], rs1160312 [11], and rs6113491 [12] SNPs with AGA and subsequent reports [18,19,20,21,22], but also indicate that other factors, i.e., high consumption of foods rich in phytochemicals, BMI, and family history, are independent risk factors for AGA among subjects bearing the variants.

The associations found in our study between AGA and BMI and family history in first-degree relatives are in agreement with a number of studies [23]. The role of smoking in AGA is controversial. Several studies reported an association between smoking and AGA development [14,24,25,26,27], while other studies showed no association [9,18,28,29,30]. In our study, smoking was associated with an increased risk in the univariate analysis, but the effect disappeared in the multivariate analysis after controlling for diet and genetic factors. Similarly, Ellis and coworkers reported that smoking status had little effect on the estimated OR of the SNP rs6152 [31], one of the first chromosome X polymorphisms near the AR locus that has been linked to baldness.

As mentioned in the introduction, AGA is linked to many chronic diseases. In our study, the presence of chronic diseases was not associated with an increased risk of AGA. As aging is a well-known contributor to the risk of chronic diseases, the observed lack of an association may trivially depend on the fact that our AGA subjects were indeed younger than the controls. On the other hand, the inclusion of older non-AGA subjects has the theoretical advantage of avoiding potential false-negative controls, since AGA is an age-dependent condition. Nonetheless, even if the ORs were adjusted for age, a residual confounding of age is not excluded.

After the adjustments for age, BMI, family history of AGA, smoking, salad consumption, and the use of fresh herbs, we found an increased risk for the TT genotype of the rs1041668 SNP present near the AR gene locus that has been associated with AGA, even if the definition of its exact role is still unknown [8]. After the adjustment, we found an increased risk also for subjects carrying the alleles A of rs1160312 and rs6113491 present on chromosome 20. Marcińska and coworkers reported that the two SNPs, rs1160312 and rs6113491, have a strong linkage disequilibrium (94%), even though both were included in their predictive test [19]. Both SNPs are present in intergenic zones, but rs1160312 is present within LINC01432, a non-coding RNA gene that was found differentially expressed in the testes of three out of seven individuals [32], where androgen biosynthesis occurs, suggesting a link between the rs1160312 polymorphism and AGA etiopathogenesis involving differential production or sensitivity to androgen levels through the modulation of LINC01432 gene activity. Further research is required to clarify how rs1160312 contributes to AGA susceptibility, starting for example from the correlation of the SNP genotype with LINC01432 expression in hair follicles.

The effects of familiarity and the protective effect of a high consumption of salad and of the use of fresh herbs remained statistically significant for all three polymorphisms, showing that the etiopathogenesis of AGA has a strong genetic component but is affected by nutrition, and thus confirming its multifactorial nature [33]. Specifically, the observed effects of fresh herb use and salad consumption are in agreement with the results of randomized clinical trials employing phytochemicals either topically [34,35] or simultaneously topically and orally [36]. Phytochemicals have antioxidant, anti-inflammatory, and anti-tumor properties that are considered responsible for the ability of plant food nutrients to prevent chronic diseases [37]. In this respect, Agaoglu et al. recognized the insufficient intake of plant foods as a risk factor for early-onset AGA in young men [26], and Bazmi and coworkers found higher dietary inflammatory index and lower antioxidant index scores in women with AGA [38]. With respect to fresh herbs, among molecules characterized by high bioactivity and included in the abovementioned trials, rosmarinic acid is noteworthy, with its properties well-known for hair health [10]. Further studies are warranted, especially clinical trials with genetic risk screening and controlled diets of at-risk subjects, in order to confirm our findings and eventually set up personalized approaches to AGA management that could include specific dietary modifications and other factors, such as, for example, the gut microbiota that has been recently linked to AGA [39].

## Figures and Tables

**Table 1 nutrients-17-00299-t001:** Characteristics and genotypes of the subjects participating in the study.

	Cases (N = 104)		Controls (N = 108)		
	N. ^a^	%	N. ^a^	%	*p*-Value ^b^
**Sex**					
male	104	100	108	100	
**Age, y**					
mean (SD)	28.5 (8.7)		38.9 (12.2)		
median (IQR)	26 (22–33)		38 (29–47)		0.0001 ^c^
**Educational level**					
up to intermediate school	9	8.7	13	12.0	
high school	58	55.8	51	47.2	
degree	37	35.6	44	40.7	0.43
**Body mass index (kg/m^2^)**					
<25	86	82.7	58	53.7	
≥25	18	17.3	50	46.3	<0.0001
**Family history of AGA (1st-degree relatives)**					
no	43	41.7	82	80.4	
yes	60	58.3	20	19.6	<0.0001
**Presence of some illness–condition ^d^**					
no	43	41.7	48	44.9	
yes	61	59.2	59	55.1	0.61
**Ever-smoking**					
never	68	65.4	56	51.9	
quit and current	36	34.6	52	48.1	0.05
**Norwood–Hamilton scale**					
mild	69	66.3	...		
moderate/severe	35	37.5	...		
**rs1160312**					
GG	25	24.0	44	40.7	
AA	27	26.0	18	16.7	
AG	52	50.0	44	40.7	
00	...		2	1.9	0.02
**rs1041668**					
TC	9	8.7	27	25.0	
TT	95	91.3	80	74.1	
00	...		1	0.9	0.001
**rs6113491**					
CC	26	25.0	45	41.7	
AC	52	50.0	44	40.7	
AA	26	25.0	19	17.6	0.03

Abbreviations: SD, standard deviation; IQR, interquartile range. ^a^: Totals may vary because of missing values. ^b^: χ^2^ or Fisher’s exact test, where appropriate. ^c^: Mann–Whitney U test. ^d^: e.g., diabetes, cardio-cerebrovascular disease, cancer, and dyslipidemia.

**Table 2 nutrients-17-00299-t002:** Association between AGA and rs1160312, rs6113491, and rs1041668 genotypes of the subjects participating in the study.

	Model 0		Model 1		Model 2	
	OR ^a^	95%CI	OR ^b^	95%CI	OR ^c^	95%CI
**rs1160312**						
GG	1		1		1	
AA AG	2.24	(1.24–4.06)	2.51	(1.20–5.25)	2.97	(1.34–6.62)
**rs1041668**						
TC	1		1		1	
TT	3.56	(1.58–8.02)	4.44	(1.66–11.9)	4.47	(1.60–12.5)
**rs6113491**						
CC	1		1		1	
AA AC	2.14	(1.19–3.85)	2.53	(1.22–5.27)	2.99	(1.37–6.52)

^a^: Model 0, crude odds ratio. ^b^: Model 1, odds ratio adjusted for age, BMI, family history of AGA, and ever-smoking. ^c^: Model 2, odds ratio adjusted for age, BMI, family history of AGA, ever-smoking, and consumption of salad and fresh herbs.

## Data Availability

The data presented in this study are available on request from the corresponding author.

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
