# Peer review of "Genetic Variants and Lifestyle Factors in Androgenetic Alopecia Patients: A Case–Control Study of Single Nucleotide Polymorphisms and Their Contribution to Baldness Risk"

_nutrients, 2025, doi:10.3390/nu17020299_

Round 1

Reviewer 1 Report

Comments and Suggestions for Authors

The title of the article is informative but could benefit from simplification to improve clarity and accessibility. A shorter title focusing on the integration of genetic and lifestyle factors in Androgenetic Alopecia (AGA) would make it more engaging. The abstract provides a good overview of the study but could better emphasize its novelty and practical implications. Including specific numerical results, such as odds ratios for the SNPs, would enhance its impact. Additionally, the abstract should clearly articulate how this study builds upon existing knowledge, particularly in controlling for nutritional and lifestyle factors alongside genetic analysis.

The introduction effectively sets the stage by providing background on AGA, its genetic basis, and its association with environmental and nutritional factors. However, it could be improved by summarizing key studies instead of listing numerous citations, which might overwhelm the reader. The choice of SNPs and their relevance to AGA should be explained more thoroughly, particularly why these specific variants were selected. Similarly, the role of the Mediterranean diet in influencing AGA risk could be elaborated upon, perhaps by comparing it to other dietary patterns or explaining its unique components that might affect hair health.

The methods section is detailed, but several areas need clarification. The rationale for selecting participants from the Lazio region should be explicitly stated, as this may limit the generalizability of the findings. The validity and reliability of the food frequency questionnaire (FFQ) used for dietary assessment should be discussed, especially in the context of its application to dermatological conditions. The genotyping procedures are adequately described, but it would be helpful to include more information about quality control measures and the reproducibility of the results. Additionally, the statistical analysis section would benefit from a justification for the inclusion of specific covariates in the logistic regression models and a discussion of how missing data were handled.

The results are presented clearly, but several areas require further attention. The significant age difference between cases and controls is notable and should be addressed as a potential limitation or confounder. The presentation of SNP associations could be enhanced with visual aids, such as graphs or charts, to make the findings more accessible. The analysis of diet and lifestyle factors is interesting but could delve deeper into which specific components of the diet had the strongest associations with AGA. Interaction terms between genetic and lifestyle factors could also be explored to provide a more comprehensive understanding.

The discussion ties the findings back to existing literature effectively but could do more to highlight the study's unique contributions. The results regarding BMI, diet, and family history as independent risk factors are compelling, yet the lack of a significant association with smoking in multivariate models should be explored further. The potential mechanisms by which these SNPs influence AGA risk, particularly rs1160312 within the LINC01432 gene, are not well addressed and could be expanded upon. Additionally, the discussion should acknowledge the limitations of the study more explicitly, such as the regional and gender-specific sample, as well as the cross-sectional design. Speculation about how these findings might inform personalized approaches to AGA management, including dietary modifications or genetic risk screening, would strengthen the practical relevance of the work.

The conclusion is succinct but should avoid overgeneralizing the findings, such as suggesting that plant-based nutrients prevent AGA. Instead, the emphasis should be on the potential protective association observed in this study. The conclusion should also propose specific directions for future research, such as longitudinal studies to explore causality or investigations into the role of gut microbiota in AGA.

Author Response

Please find the response in the attached file

Reviewer 2 Report

Comments and Suggestions for Authors

Roberto Ambra et al. reported an interesting paper about Androgenetic Alopecia. The submission is a Communication, which reported preliminary results. The topic fell within the scope of Nutrients. The reviewer suggested a Minor Revision for this paper. Please refer to the following comments:

1)      Please define the abbreviations of TC, TT, etc., in Table 1 and 2.

2)      The authors stated that “while smoking was not statistically significant anymore (data not shown)”. It would be better to show the data.

3)      In the Discussion Section, the comparison of this study with current clinical trials could be added.

4)      The writing style of Conclusion Section was improper. The current content was more similar to a Discussion. Please combine it with the previous Discussion Section, and rewrite a Conclusion Section.

Author Response

(The authors gave the same response as above.)

Round 2

Reviewer 1 Report

Comments and Suggestions for Authors

Thank you for the revision which I find satisfactory.